# Administration of Human MSC-Derived Extracellular Vesicles for the Treatment of Primary Sclerosing Cholangitis: Preclinical Data in MDR2 Knockout Mice

**DOI:** 10.3390/ijms21228874

**Published:** 2020-11-23

**Authors:** Roberta Angioni, Bianca Calì, Vasanthy Vigneswara, Marika Crescenzi, Ana Merino, Ricardo Sánchez-Rodríguez, Cristina Liboni, Martin J. Hoogduijn, Philip Noel Newsome, Maurizio Muraca, Francesco Paolo Russo, Antonella Viola

**Affiliations:** 1Department of Biomedical Sciences, University of Padova and Fondazione Istituto di Ricerca Pediatrica—Città della Speranza, 35127 Padova, Italy; roberta.angioni@unipd.it (R.A.); bianca.cali@ior.usi.ch (B.C.); ricardo.sanchezrodriguez@unipd.it (R.S.-R.); cristina.liboni@phd.unipd.it (C.L.); 2National Institute for Health Research Biomedical Research Centre at University Hospitals Birmingham NHS Foundation Trust and the University of Birmingham; Centre for Liver and GI Research, Institute of Immunology and Immunotherapy, University of Birmingham; Liver Unit, University Hospitals Birmingham NHS Foundation Trust, Birmingham B15 2TH, UK; V.Vigneswara@bham.ac.uk (V.V.); p.n.newsome@bham.ac.uk (P.N.N.); 3Department of Surgery, Oncology and Gastroenterology—DiSCOG, Gastroenterology and Multivisceral Transplant Unit, 35128 Padova, Italy; marikacrescenzi@hotmail.com (M.C.); francescopaolo.russo@unipd.it (F.P.R.); 4Nephrology and Transplantation, Department of Internal Medicine, Erasmus Medical Center, 3015 CN Rotterdam, The Netherlands; a.merinorodriguez@erasmusmc.nl (A.M.); m.hoogduijn@erasmusmc.nl (M.J.H.); 5Fondazione Istituto di Ricerca Pediatrica Città della Speranza, Padova, and Stem Cell and Regenerative Medicine Laboratory, Department of Women’s and Children’s Health, University of Padova, 35128 Padova, Italy; muraca@unipd.it

**Keywords:** inflammation, fibrosis, mesenchymal stromal cells, liver, therapy

## Abstract

Primary Sclerosing Cholangitis (PSC) is a progressive liver disease for which there is no effective medical therapy. PSC belongs to the family of immune-mediated biliary disorders and it is characterized by persistent biliary inflammation and fibrosis. Here, we explored the possibility of using extracellular vesicles (EVs) derived from human, bone marrow mesenchymal stromal cells (MSCs) to target liver inflammation and reduce fibrosis in a mouse model of PSC. Five-week-old male FVB.129P2-Abcb^4tm1Bor^ mice were intraperitoneally injected with either 100 µL of EVs (± 9.1 × 10^9^ particles/mL) or PBS, once a week, for three consecutive weeks. One week after the last injection, mice were sacrificed and liver and blood collected for flow cytometry analysis and transaminase quantification. In FVB.129P2-Abcb4^tm1Bor^ mice, EV administration resulted in reduced serum levels of alkaline phosphatase (ALP), bile acid (BA), and alanine aminotransferase (ALT), as well as in decreased liver fibrosis. Mechanistically, we observed that EVs reduce liver accumulation of both granulocytes and T cells and dampen VCAM-1 expression. Further analysis revealed that the therapeutic effect of EVs is accompanied by the inhibition of NFkB activation in proximity of the portal triad. Our pre-clinical experiments suggest that EVs isolated from MSCs may represent an effective therapeutic strategy to treat patients suffering from PSC.

## 1. Introduction

Primary sclerosing cholangitis (PSC) is an immune-mediated, cholestatic liver disease, characterized by persistent, progressive, biliary inflammation and fibrosis. There is no effective medical therapy for this condition, and thus, patients commonly progress to end-stage liver disease necessitating liver transplantation [1,2]. The incidence of PSC ranges from 1/446,000-1/6, 170 inhabitants/year with a male to female ratio of approximately 2:1. In Europe, cholestatic liver disease, as a whole, accounted for more than 10% of all liver transplant activity. The cause of PSC is unknown, however, genetic abnormalities of the immune regulation, viral infection, toxins from intestinal bacteria, bacteria in the portal venous system, ischemic vascular damage, and toxic bile acids from intestinal bacteria have all been implicated in its pathogenesis [3]. Currently, genetic and immunological factors are critical in its pathogenesis, due to the familial occurrence of PSC and its association with HLA-B8, DR3, DR2, and DR4. Evidence of abnormal immuno-regulation is further highlighted by infiltration of the bile ducts with lymphocytes, increased serum gamma globulins, increased circulating immune complexes, and increased metabolism of complement C3 [4]. Therefore, the biliary injury in PSC is a reflection of activation and recruitment of immune cells that drive progressive biliary fibrosis resulting in end-stage liver failure [5]. PSC is associated with a strong liver infiltration of innate immune cells. Although the inflammatory infiltrate of PSC is largely comprised of T cells [5], other cell types, including macrophages and granulocytes, play important roles in the immunopathogenesis of PSC, sustaining chronic inflammation and triggering periductal fibrosis. As a consequence of this, patients show altered cholestasis, obstructive structures, and eventually biliary cirrhosis [5].

Due to their beneficial role in immunomodulation and tissue repair, MSCs have been exploited in several clinical trials for the treatment of numerous inflammatory disorders, such as graft versus host disease (GVHD), diabetes, lung fibrosis, heart failure and Crohn’s disease [6]. Despite significant heterogeneity among previous studies, a single administration of bone marrow MSC has been demonstrated to improve liver function in patients with liver disease [7]. Among the mechanisms involved in the therapeutic efficacy of transplanted MSC, a major role is played by the release of extracellular vesicles (EVs) [8,9]. EVs are cell-derived membrane particles, whose size ranges from 30 nm up to several microns, characterized by distinctive molecular composition depending on their origin, size and content [10]. EVs have attracted increasing interest as therapeutic tools and it has been recently shown that EVs from liver stem cells (LSCEVs) are able to reduce ductular reaction and biliary fibrosis in MDR2^−/−^ mice [11].

In this study, we explored the possibility of using EVs derived from human mesenchymal stromal cells (MSCs) to target liver inflammation and reduce fibrosis in a mouse model of PSC.

## 2. Results

### 2.1. MSC-Derived EVs Ameliorate the Clinical Outcome in a Mouse Model of Human PSC

FVB.129P2-Abcb4^tm1Bor^ mice (FVB.Mdr2^−/−^ from here forward) are characterized by the complete inability of the liver to secrete phospholipids into the bile, and thus, spontaneously develop progressive chronic biliary injury and fibrosis, providing a valid model of human PSC [12,13,14]. EVs were isolated from bone marrow-derived human MSC by ultrafiltration, using Amicon^®^ Ultra 15 mL Filters (Merck Millipore) following manufacturer’s instructions, as graphically represented in Figure 1A. In details, medium conditioned by MSCs (MSC-CM) was collected and loaded into Amicon^®^ Ultra 15 mL Filters to proceed with EV isolation by ultrafiltration. Then, validation of the EV isolation procedure was performed by measuring the size of the purified particles by Nanosight (Figure 1B) and by visualizing them with TEM (Figure 1C). Both techniques revealed the presence of a heterogenous preparation of vesicles. The size of the particles ranged from 45 to 372 nm, and the mode size was 78.1 ± 4.8 nm. The average number of particles generated from 10^6^ MSCs was 9.1 × 10^9^ particles/mL. The western blot analysis of the conventional EV markers, CD63 and CD9 [15,16], confirmed the accuracy of the exploited EV isolation procedure (Figure 1D). To assess the therapeutic potential of human MSC-derived EVs in models of PSC, 5-week-old male FVB.Mdr2^−/−^ mice were treated with multiple i.p. injections of EVs (100 µL of EVs ± 9.1 × 10^9^ particles/mL) or vehicle (100 µL PBS without calcium and magnesium), as depicted in Figure 1E. After 3 weeks of treatment with EVs or PBS, serum levels of hepatic damage factors, in particular alanine aminotransferase—ALT (Figure 1F), alkaline phosphatase—ALP (Figure 1G) and biliary acid—BA (Figure 1H) were analyzed. FVB.Mdr2^−/−^ mice treated with EVs had significantly lower levels of serum ALT, ALP and BA compared with 8-week-old male untreated mice.

### 2.2. EVs Have Anti-Fibrotic Properties in the FVB.Mdr2^−/−^ Mouse Model

Biliary fibrosis is a hallmark feature of PSC in both humans and FVB.Mdr2^−/−^ mice, with subsequent loss of hepatic function [17,18]. Therefore, livers from mice were analyzed to evaluate the effect of EVs on the fibrotic process. As expected [19,20], livers from FVB.Mdr2^−/−^ mice display consistent fibrosis, measured by Sirius red staining (Figure 2A, upper panel). However, the administration of EVs resulted in a reduction in Sirius red positive areas (Figure 2A,B). Consistently, immunohistochemical analysis of alpha-smooth muscle actin (alpha-SMA), a marker of myofibroblast differentiation [21] and liver fibrosis [22], demonstrated that administration of EVs reduced expression of alpha-SMA (Figure 2C,D).

### 2.3. EVs Reduce Liver Immune Infiltration in the FVB.Mdr2^−/−^ Mouse Model 

We analyzed the effects of EV administration on liver infiltrate of FVB.Mdr2^−/−^ mice. EV administration decreased the total amount of CD45+ immune cells in the liver (Figure 3A), as well as numbers of lymphoid and myeloid cells, as indicated by the analysis of CD3+ and CD11b+ populations, respectively (Figure 3B,E). In particular, among CD3+ cells, EV administration significantly reduced the CD8+ T cell population, whilst leaving CD4+ T cells unaffected (Figure 3C, D). EV administration also reduced granulocyte infiltration (CD11b+, Ly6C+ and Ly6G+ cells), but had no significant effect on the percentage of liver macrophages (CD11b+, F4/80+ cells) (Figure 3F,G). Taken together, these results highlight the effect of MSC-derived EVs in dampening liver inflammation, associated with PSC. However, despite several studies pointed out a central role of macrophages in the immunosuppressive effect of MSC-derived EVs [23,24], our multiparametric flow cytometry analysis indicated that hEVs do not affect macrophage accumulation in MDR2^−/−^ mouse liver (Figure 3G). The results were further corroborated by IHC analysis (Appendix A). Also, with the aim of investigating whether hEVs were able to alter macrophage polarization, we induced, MDR2^-/-^ bone-marrow derived macrophage polarization to M0, M1 or M2 phenotypes either in the presence or absence of hEVs for 24 h. The transcript analysis of M1 (NOS2, IL1b and IL6, Appendix A) and M2 (ARG, MRC and Fizz, Appendix A) associated -genes indicated that hEVs did not interfere with macrophage polarization either in MDR2^+/+^ and in FVB.Mdr2^−/−^.

### 2.4. EVs Reduce Liver Expression of VCAM1 in the FVB.Mdr2^−/−^ Mice 

The presence of adhesion molecules in portal spaces, as the expression of vascular cell adhesion molecule- 1 (VCAM-1), is an indicator of hepatic high permeability thus reflecting an important influx of immune cells. To better understand the mechanism by which administration of MSC-EVs reduces immune cell infiltration into livers, we analyzed VCAM-1 expression in liver sections of FVB.Mdr2−/− mice treated with EVs. Image analysis showed a remarkable reduction of VCAM-1 positive cells in liver sections of mice that have been treated with EV, compared to control ones (Figure 4A,B). To note, nuclear factor-kappa B (NFκB) plays a prominent role in rapid induction of VCAM-1 expression by inflammatory mediators [25]. As phosphorylation of p65 subunit is required for nuclear translocation and transcriptional activation [26], we analyzed by immunohistochemistry the levels of phospho-p65 in paraffinized liver sections from FVB.Mdr2−/− mice treated with EVs, using mice injected with PBS as control. EV administration significantly reduced phospho-p65 immunohistochemical expression in proximity of the portal triad, indicating that EVs reduces NFκB activation in PSC livers (Figure 4C,D).

## 3. Discussion

An effective therapy for biliary disorders and, in particular, for PSC is still missing, and this study demonstrates that EVs isolated from human bone marrow-derived MSCs reduce biliary inflammation and fibrosis, suggesting that EVs have a therapeutic potential in the control of PSC disease progression. The powerful regulatory role of MSCs in both adaptive and innate immune responses has led to extensive exploitation of MSCs in clinical trials as immunosuppressive agents for autoimmune and inflammatory diseases, including graft-versus-host disease (GVHD), multiple sclerosis (MS) and systemic lupus erythematosus (SLE) [27,28]. Notably, therapeutic strategies using MSCs offer promise for the treatment of liver diseases [29], with the therapeutic effect being attributed mainly to their immunosuppressive actions and/or release of trophic factors, which are involved in tissue regeneration, induction of angiogenesis, control of apoptosis and fibrosis.

We previously demonstrated that MSCs do not require homing to specific organs [30] and that paracrine signals, mediated by secreted proteins, inhibit high endothelial cell activation and immune response during acute inflammation [31]. In addition, numerous studies on immune cells have already demonstrated that MSC-derived EVs are able to mimic the effects of the cell of origin [32]. EVs are multi-signal messengers that have a key role in intercellular signalling by carrying cargo, such as mRNA, miRNA, and proteins [33,34]. EVs are present in multiple biological fluids, including the bile, and may represent a relevant vehicle for new therapeutic approaches for biliary disorders, due to interactions between cholangiocytes and their surrounding tissues [35]. Interestingly, it has been recently shown that EVs from liver stem cells (LSCEVs) are able to reduce ductular reaction and biliary fibrosis in MDR2^−/−^ mice [11].

Here we demonstrated that the administration of EVs derived from MSCs inhibit PSC progression in the MDR2^−/−^ mouse model, by reducing peribiliary fibrosis and inflammation.

T-cell–mediated biliary injury is a feature of PSC and, in our analysis, we found that MSC-derived EVs prevent the accumulation of liver-infiltrating cytotoxic CD8+ T cells in FVB.Mdr2−/− mice. Notably, we also found a remarkable reduction in liver-infiltrating granulocytes, which are known to exacerbate liver damage. Neutrophil accumulation and activation have been implicated in the pathogenesis of organ damage [36], especially liver injury, and indeed neutrophil accumulation is also a prominent feature of non-alcoholic steatohepatitis (NASH). Moreover, a high neutrophil-to-lymphocyte ratio (NLR) is typical of patients with advanced liver disease and fibrosis [37,38,39,40]. Chronic liver diseases, including PSC, are characterized by continuous liver cell injury followed by compensatory proliferation and deposition of extracellular matrix proteins from activated hepatic stellate cells (HSCs), which transform into liver myofibroblasts during inflammation [41]. HSC proliferation and differentiation are mainly induced by TGF-β1, Platelet-derived growth factor-B (PDGF-B), and Angiotensin II [42,43], but reactive oxygen species (ROS) produced by neutrophils also play a major role in the process [44,45]. Here, we show that EV administration reduces liver expression of alpha-SMA, a marker of myofibroblast differentiation, and concomitantly affects granulocyte infiltration. These findings are in line with previous studies supporting the existence of a cross-talk between neutrophils and HSC. It has been in fact reported that neutrophils are required for HSC activation and that, in turn, activated HSCs prolong neutrophil survival by producing granulocyte-macrophage colony-stimulating factor and interleukin-15 [45], eventually sustaining a positive forward loop to promote liver damage and fibrosis. 

The vascular architecture of the liver predisposes it to be a trapping site for leukocytes [46]. In the liver, the low blood flow rates through sinusoidal spaces makes the requirement for selectin-mediated rolling and tethering less relevant. This is particularly clear in the case of CD8+ T cells and neutrophils, which accumulate at normal rates in the livers of E and P selectin-deficient mice [47,48]. Therefore, the integrins ICAM-1, VCAM-1, and vascular adhesion protein-1 (VAP-1) are likely to be responsible for the trapping mechanism of these immune cells [49]. Although ICAM-1 is constitutively expressed on hepatocytes, liver sinusoidal endothelial cells (LSECs), and Kupffer cells, VCAM-1 expression is mainly induced in LSECs under inflammatory conditions [50,51]. Interestingly, binding of lymphocytes to hepatic endothelium could be blocked in vitro by anti-VCAM-1 Abs [52]. Moreover, it has been suggested that whilst ICAM-1-mediated adhesion requires local antigen presentation, VCAM-1-mediated trapping of CD8+ T cells within the liver does not require local antigen availability. As for neutrophils, although the specific signals regulating liver infiltration in PSC have not been investigated, VCAM-1 has been shown to specifically mediate their adhesion to endothelium in sepsis patients [53], suggesting that under inflammatory conditions the alpha-4-integrin/VCAM-1 pathway may be selectively modulated to allow neutrophil recruitment into target organs. We previously showed that, in a simple model of local inflammation, mouse MSCs inhibit endothelial cell activation, affecting the expression of both VCAM-1 and ICAM-1 in draining lymph node (LN) endothelial vessels [31]. In the same line, in the present study, we show that EVs derived from human MSC are effective in reducing VCAM-1 expression in liver, thus suggesting that the observed reduction in immune cell infiltration is the result of this anti-inflammatory effect. NFkB plays a crucial role in the regulation of VCAM1 expression in the liver [54,55] and deregulation in the NfkB signaling during the PSC pathophysiology has been already reported [56]. In addition, previous studies have reported specific targeting of NFkB by MSCs [57]. Therefore, we speculated that a direct involvement of NFkB inhibition, mediated by human EVs, arises in the anti-inflammatory effects that we observed. We found that human EVs are able to inhibit the activation of NFkB, evaluated in term of its phosphorylation. This effect is markedly evident in proximity of the hepatic triad, where the leukocytes recruitment usually occurs. However, the molecular mechanism responsible for this anti-inflammatory effect requires deeper investigation. 

In conclusion, these data demonstrate that EVs, isolated from human MSCs, may represent an effective therapeutic strategy in treating patients with PSC. Most importantly, our preclinical experiments support further development of MSC-derived EVs for the treatment of PSC, for which current major indication is liver transplantation. Notably, a cumulative effort is required to regulate the use of EVs in therapy and overcome the current limitations in manufacturing levels, such as the production of an appropriate amount of GMP-quality EVs, the optimization of the EV characterization procedure (definition of identity and purity) and the identification of optimal storage and distribution conditions.

## 4. Materials and Methods

### 4.1. Isolation of Human MSCs

Human, bone marrow-derived. MSCs were provided by Orbsen Therapeutics Ltd. (Galway, Ireland). Ethical approvals were granted from the NUI Research Ethics Committee (REC) and the Galway University Hospitals Clinical Research Ethics Committee (CREC). Briefly, bone marrow was harvested from volunteers, and cell culture was set up as previously described [58]. MSCs were characterized according to international guidelines [59]. All samples were obtained with informed consent. Procurement of the sample conformed to European Parliament and Council directives (2001/20/EC; 2004/23/EC).

### 4.2. Isolation and Characterization of Extracellular Vesicles (EVs)

Collection of MSC-CM was performed as described [60]. EVs were isolated and characterized in compliance with the MISEV2018 guidelines [61] EVs were isolated from MSC-CM by ultrafiltration using Amicon^®^ Ultra 15 mL Filters, Ultracel-PL PLHK, 100kDa (Merck Millipore, Darmstadt, Germany), according to the manufacturer’s instructions. EVs were collected, concentrated in 150 μL of PBS, and stored immediately at −80 °C. Analysis of absolute size distribution and concentration of EVs was performed using NanoSight NS300 (NanoSight Ltd., Lyon, France), equipped with a 405-nm laser and a highly sensitive digital camera. With NTA, particles were tracked and sized based on Brownian motion and the diffusion coefficient. Samples were diluted in filtered PBS (0.2 µm pore size) to obtain the right number of particles (1 × 10^8^ particles/mL), in accordance with the manufacturer’s recommendations. The NTA measurement conditions were: detection threshold 3, temperature 23.61 ± 0.8 °C; viscosity 0.92 ± 0.02 cP, 25 frames per second. For each sample, three videos of 30 s duration were obtained with 10 s delay between recordings, generating three replicate histograms. Each video was analyzed to give the mean, mode, and estimated concentration for each particle size. Total proteins from purified EVs were extracted with PBS 0.4% SDS and quantified by MicroBCA kit (Pierce, Thermo Fisher Scientific, Waltham, MA, USA). Three µg of proteins were separated by 10%SDS-PAGE under non-reductive conditions for CD9 (Biolegend, San Diego, CA, USA) and CD63 (Abcam, Cambridge, UK). Chemiluminescence was achieved by ECL Prime WB Detection reagent (GE). Images were acquired with ImageQuant LAS 4000Mini (GE), following the manufacturer’s instructions. For the transmission electron microscopy, 20 µL of EVs were dispensed for 2 min on 300 mesh carbon-coated copper grids that were made hydrophilic by a 15-s exposure to a glow discharge. A filterpaper (Whatman, Maidstone, UK) was used to remove the excess of liquid. EVs were then stained with 1% uranyl acetate for 2 min. FEI Tecnai G2 transmission electron microscope operating at 100 kV, with a Veleta (Olympus Soft Imaging System, Münster, Germany) digital camera, was exploited to capture images.

### 4.3. Animal Experimentation 

All animals received humane care according to the criteria outlined in the “Guide for the Care and Use of Laboratory Animals” prepared by the National Academy of Sciences and published by the National Institutes of Health (NIH publication 86-23 revised 1985). The experiments were approved by the Italian Ministry of Health under the legislative Decrees no 332/2013-B and no 88/2017-PR. 

### 4.4. In Vivo Treatments

Five-week-old male FVB.129P2-Abcb4tm1Bor mice were intraperitoneally injected with either 100 µl of EVs (± 9.1 × 10^9^ particles/mL) or PBS, once a week for three consecutive weeks. One week after the last injection, mice were sacrificed and liver and blood collected for FACS analysis and transaminase quantification, respectively. To assess liver injury, ALT and ALP activity and also BA levels were measured in serum samples at the clinical biochemistry laboratory of the Women’s Hospital Birmingham (Birmingham, UK).

### 4.5. FACS Analysis of Liver Infiltrate 

Mice were euthanized by cervical dislocation. A needle was inserted into the portal vein and liver perfused with 20 mL pH 7.0 PBS before being harvested. Livers were enzymatically digested with 0.2 mg/mL DNAse I (18068015 Thermo Fisher Scientific, Waltham, MA USA) and Collagenase type II (17101015 Thermo Fisher Scientific, Waltham, MA USA), in RPMI 0.5% BSA, for 60 min at 37 °C and the resulting immune cells were then purified and analyzed by flow cytometry. Inflammatory cells were gated as a CD45+ cell population (PerCP-Cy5.5, clone 30-F11; BD Biosciences, San Jose, CA, USA), and nonviable cells were excluded using a LIVE/DEAD fixable stain (Invitrogen, Thermo Fisher Scientific, Waltham, MA, USA). Lymphocytes were characterized based on staining using a cocktail of anti–CD3 (FITC, clone 145-2C11; eBioscience, Thermo Fisher Scientific, Waltham, MA, USA); anti–CD4 (PE, clone V4; BD Biosciences, San Jose, CA, USA); anti–CD8a (eFluor 450, clone 53-6.7; eBiosciences, Thermo Fisher Scientific, Waltham, MA, USA). Myeloid cell subsets were identified by staining with anti–CD11b (PerCP-Cy5.5, clone M1/70; BD Biosciences, San Jose, CA, USA); anti–Ly6G (APC, clone RB6-8C5; eBiosciences, Thermo Fisher Scientific, Waltham, MA, USA); and anti–F4/80-FITC (clone BM8; eBiosciences, Thermo Fisher Scientific, Waltham, MA, USA). Samples were examined with a BD FACSCanto II (BD Biosciences, San Jose, CA, USA) and data were analyzed using FlowJo software (BD Biosciences, San Jose, CA, USA).

### 4.6. Immunohistochemical Analyses

After deparaffinization and antigen unmasking, sections were incubated overnight at 4 °C with one of the following primary antibodies: mouse anti-mouse alpha smooth muscle actin (α-SMA, clone 1A4, Sigma Aldrich, Merck Millipore, Darmstadt, Germany) diluted 1:2000 in PBS, phospho-p65 (27.Ser 536) (sc-136548, Santa Cruz Biotechnology, Inc., Dallas, TX, USA) diluted 1:100 in PBS. After incubation with the corresponding biotinylated secondary antibody (goat anti-mouse IgG H+L biotinylated #ab6788 2000 mg/mL, Abcam, Cambridge, UK), the enzymatic reaction was developed for 4 min, using diaminobenzidine (DAB) solution as substrate (Vector Laboratories, Maravai LifeSciences, San Francisco Bay Area, CA, USA). Nuclei were counterstained with hematoxylin (Diapath, Bergamo, Italy). Signal quantification was performed as follows; for each sample the positive areas were calculated using ImageJ software and the data were normalized as specified in figure legend. For fibrosis analysis, Sirius red staining protocol was performed. Once deparaffinization and hydration of liver sections was undertaken, nuclei were stained with Mayer’s haematoxylin. Then, Sirius red solution (Direct Red 80, 365548 Sigma-Aldrich Merck Millipore, Darmstadt, Germany) in 1.3% saturated aqueous solution of picric acid) was used for the labelling of collagen (1hour at room temperature). Slices were washed in acidified water (0.5% acetic acid in water), dehydrated and mounted. Signal quantification was performed as follows: for each sample, 6 fields (×10) were scored using a Leica microscope; the positive areas were calculated using ImageJ software and the data were normalized as specified in figure legend.

### 4.7. Bone Marrow Derived Macrophages (BMDM) and RT-PCR Real Time

BMDM were obtained by flushing from murine FVB.129P2-Abcb4tm1Bor femur and tibia (or WT animals as control) and differentiated into macrophages in RPMI 1640-10% FBS in the presence of M-CSF (40 ng/mL) for 5 days and then refill with 2 mL more with M-CSF (20 ng/mL) for 2 days more, at day 7 cells were used to differentiated to M1 with LPS (500 ng/mL) and IFNγ (25 ng/mL) or M2 with IL-4 (25 ng/mL) in the presence or absence of EVs (± 9.1 × 10^8^ particles/mL). Total RNA was extracted by Trizol (Invitrogen, Thermo Fisher Scientific, Waltham, MA, USA) following the manufacturing protocol. The cDNA reactions were prepared from 500 ng of total RNA using the High Capacity cDNA Reverse Transcription Kit (Applied Biosystems, Thermo Fisher Scientific, Waltham, MA, USA), and a 1:10 dilution was used for quantitative PCR. The reactions were carried out using specific primers by SYBR Green in Quantum Real Time PCR system (Applied Biosystem, Thermo Fisher Scientific, Waltham, MA, USA). Data were normalized against the Rplp0 gene expression using the comparative ΔΔCt method. PRIMERS Sequences: Nos2 (F:GCACATTTGGGAATGGAGACTG, R:GGCCAAACACAGCATACCTGA), IL1b (F:GGACATGAGCACCTTCTTTTCC, R:TTGTTCATCTCGGAGCCTGTAG), IL6 (F:AGGATACCACTCCCAACAGAC, R:GCCATTGCACAACTCTTTTCTC), Arg1 (F:ACAAGACAGGGCTCCTTTCAG, R:GGCTTATGGTTACCCTCCCG), Mrc1 (F:TTGCACTTTGAGGGAAGCGA, R:CCTTGCCTGATGCCAGGTTA), Fizz (F: CCTGCTGGGATGACTGCTAC, R: CAGTGGTCCAGTCAACGAGT), Rplp0 (F: GGGCATCACCACGAAAATCTC, R: CTGCCGTTGTCAAACACCT).

### 4.8. Statistical Analysis

Data were analyzed using the Prism Software (GraphPad, La Jolla, CA, USA). Investigators were mostly blinded to sample allocation during experiments. In general, statistical comparison was carried out using the Student’s *t*-test, assuming normal distribution. In figures, asterisks * indicate *p* ≤ 0.05.

## Figures and Tables

**Figure 1 ijms-21-08874-f001:**
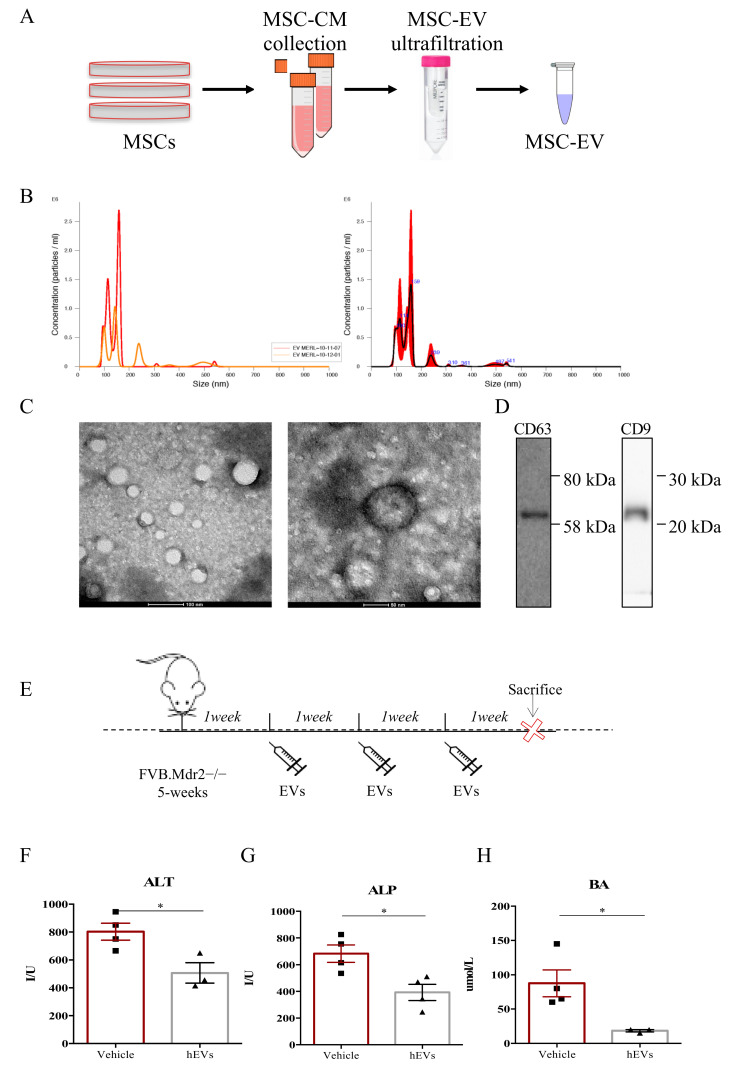
MSC-derived EVs reduce liver damage in FVB.Mdr2^−/−^ mice. (**A**) Illustration of the purification protocol of EVs from human MSCs. (**B**) Analysis of absolute size distribution and concentration of EVs was performed using NanoSight NS300 (NanoSight Ltd.), equipped with a 405-nm laser and a highly sensitive digital camera. With NTA, particles were tracked and sized based on Brownian motion and the diffusion coefficient. (**C**) Representative pictures of EVs obtained with electron microscopy. Scale bar 100 nm and 50 nm. (**D**) Western Blot for CD63 and CD9 on MSC-derived EVs. (**E**) Schematic illustration of the experimental protocol designed to investigate the effect of human MSC-derived EVs (EVs) in Mdr2^−/−^ mouse model. 5-week-old male FVB.Mdr2^−/−^ mice were intraperitoneally injected with ±9.1 × 10^8^ EVs once a week for three consecutive weeks, using PBS as vehicle control. Mice were sacrificed at nine weeks of age. (**F-H**) Serum biochemistry of ALT (**F**), ALP (**G**) and BA (**H**) in FVB.MDR2^−/−^ mice, treated with EVs or vehicle. Values are presented as means ± s.e.m., each dot refers to one mouse. * *p* < 0.05, Student’s *t*-test. ALT, alanine aminotransferase; BA, bile acid; ALP, alkaline phosphatase.

**Figure 2 ijms-21-08874-f002:**
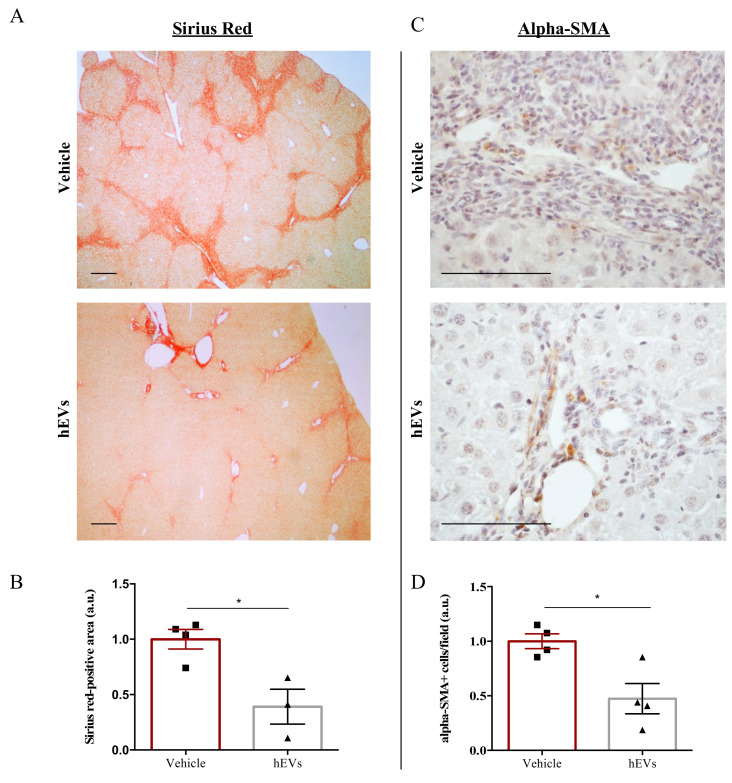
EV administration reduces liver fibrosis in FVB.Mdr2^−/−^ mice. (**A**) Representative images of connective tissue stain (Sirius red) in liver sections from FVB.Mdr2^−/−^ mice treated or not with EVs. Scale bar, 250 μm. (**B**) Quantification of liver fibrosis in FVB.Mdr2^−/−^ mice injected with EVs relative to vehicle-treated mice. Graph shows percentage of fibrotic areas (Sirius red-positive areas) in FVB.Mdr2^−/−^ liver sections, normalized to vehicle-treated mice. Bars represent means ± s.e.m from at least 4 liver sections per animal (*n* = 4 animals for vehicle, *n* = 3 animals for hEV treatment). * *p* < 0.05, Student’s *t*-test. (**C**) Representative images of α-SMA immunoreactivity in liver sections from FVB.Mdr2^−/−^ mice treated or not with EVs. Scale bar, 100 μm. (**D**) Quantification of α-SMA immunoreactivity in FVB.Mdr2^−/−^ mice injected with EVs relative to vehicle-treated mice. Bars represent means ± s.e.m from at least 5 liver sections per animal (*n* = 4 animals for vehicle, *n* = 4 animals for hEV treatment). * *p* < 0.05, Student’s t-test.

**Figure 3 ijms-21-08874-f003:**
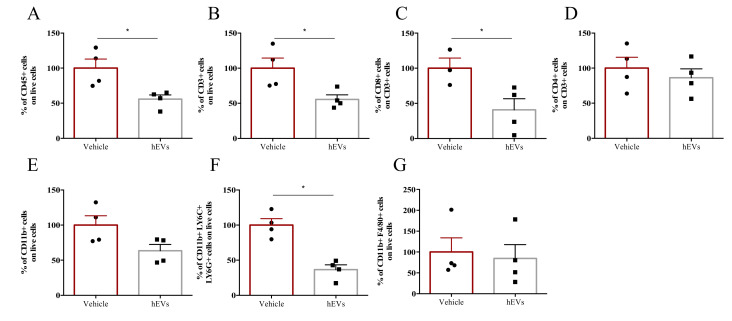
EVs dampen liver inflammation associated to PSC. Liver-infiltrating immune cells in FVB.Mdr2^−/−^ mice treated with EVs or vehicle. Percentages of CD45+ leukocytes (**A**), CD3+ lymphocytes (**B**), CD11b+ myeloid cells (**E**), CD3+CD8+ (**C**) and CD3+CD4+ (**D**) T cells, granulocytes (CD11b+, Ly6G+, Ly6C+) (**F**) and macrophages (CD11b+, F4/80+) (**G**) infiltrating livers were analyzed by flow cytometry. Data are expressed as normalized on PBS injected (Vehicle) mice (100%). Bars represent means ± s.e.m from 4 animals per group. * *p* < 0.05, Student’s *t*-test.

**Figure 4 ijms-21-08874-f004:**
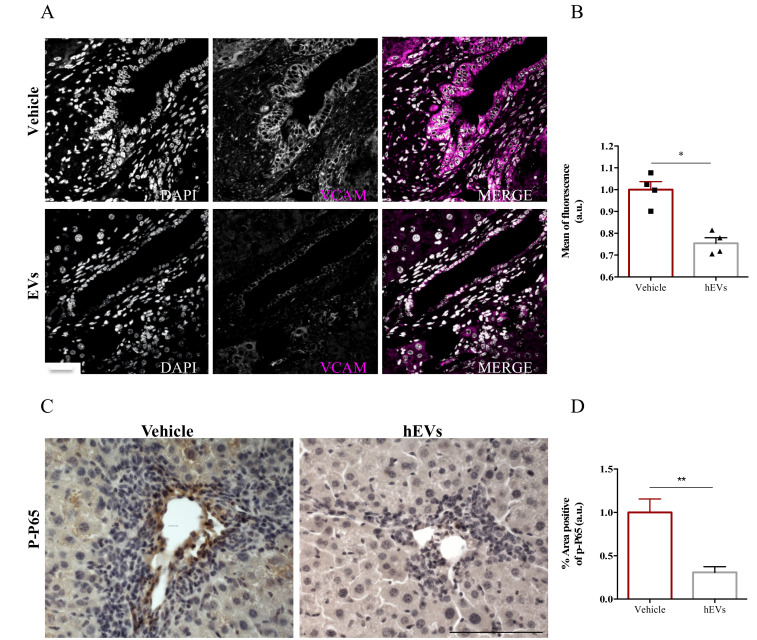
EV administration reduces VCAM-1 expression by the inflamed liver endothelium. (**A**) Representative images of VCAM-1 immunostaining (magenta) in liver sections from FVB.Mdr2^−/−^ mice treated or not with EVs. Nuclei were counterstained with DAPI (grey). Scale bar, 100 μm. (**B**) Relative quantification of VCAM1 immunoreactivity showing mean of fluorescence intensity normalized on vehicle-treated mice. Bars represent mean ± s.e.m from at least five sections from four animals per group. * *p* < 0.05, Student’s *t*-test. (**C**) Representative pictures of phospho-p65 immunostaining in liver sections from FVB.Mdr2^−/−^ mice treated or not with EVs. Nuclei were counterstained with ematoxilin. Scale bar, 100 μm. (**D**) Relative quantification of phospho-p65 immunoreactivity showing percentage of positive area normalized on vehicle-treated mice. Bars represent mean ± s.e.m from at least five sections from four animals per group. * *p* < 0.05, ** *p* < 0.005, Student’s *t*-test.

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
