# Peer review of "Administration of Human MSC-Derived Extracellular Vesicles for the Treatment of Primary Sclerosing Cholangitis: Preclinical Data in MDR2 Knockout Mice"

_ijms, 2020, doi:10.3390/ijms21228874_

Round 1

Reviewer 1 Report

Administration of Extracellular Vesicles for the 3 treatment of Primary Sclerosing Cholangitis: 4 preclinical data in MDR2 knockout mice by Angioni R et al.

The authors employed extracellular vesicles from human bone marrow mesenchymal stromal cells (MSCs) to treat Primary Sclerosing Cholangitis (PSC) in a mouse model.  EV administration improved liver function and decreased liver fibrosis. In addition, EVs reduced liver infiltration with immune cells.  Their data demonstrate that EVs isolated from human MSCs may be used to treat patients with PSC whose major indication would be liver transplantation.

This is a very interesting article highlighting the translational potential of EVs derived from MSCs. The data are convincing; the manuscript is very well written, easy to read and follow.

Few minor comments:

Please use “Sirius red” spelling consistently throughout the manuscript.

Page 5, line 137: “…that hEVs do not affect their accumulation..”

Page 6, line 168: Figure 4 legend: “..DAPI..”

Page 10, line 336: please write full name of BMDM at first mention

Reviewer 2 Report

Title: Administration of Extracellular Vesicles for the 2 treatment of Primary Sclerosing Cholangitis: 3 preclinical data in MDR2 knockout mice

Angioni et al., in the manuscript entitled “Administration of Extracellular Vesicles for the treatment of Primary Sclerosing Cholangitis: preclinical data in MDR2 knockout mice”, aimed at testing the effect of extracellular vesicles from MSCs on a mice model of PSC.

In my opinion, the results are very interesting for further evaluation and the manuscript is well organized.

Below are my comments before considering the manuscript for publication in IJMS: 

  • Authors should be more accurate in the description of the results in paragraph 2.1; for example, for figures 1 A, B, C, authors should include result description on Nanosight analysis, TEM, and WB. Some of this information is included in method section 4.2 (lines 279-281) where is not necessary to comment on the results but only the method description is needed.
  • Also, from the Nanosight results (figure 1B), the EV population seems heterogeneous compared with the results from TEM analysis. The authors should comment on that.
  • In supplementary figures 1C and D, how many samples for conditions were analyzed? From the histograms it seems two for each condition; the sample number should be increased to have more accurate results.
  • The authors didn’t clearly assess how many mice they treated in their experiments, both in Results, both in Materials and Methods sections, even the number seems to be 4 or 5 from some Figure legends. Also, in Results, they wrote that they observed differences between mice treated with EVs and “sex and age matched untreated mice” (lines 91-92), but before they have spoken about “5-week-old male FVB-Mdr2-/- mice” (line 86) as in Materials and Methods (line 300). In Figures 2B and 3C, it seems that a point (triangle/circle) is missing and in Fig 4D there are no points regarding the number of mice.
  • The molecular mechanisms should be deeply investigated in in vitro models
  • In the Discussion section, authors should include a paragraph discussing, in addition to the opportunities, also the limits that exist today for the use of EVs in therapy. For example, what is the regulatory framework in this regard? Which are the opportunities to use EVs in therapy at the manufacturing levels (large-scale purification-GMP)?
  • A brief Conclusion paragraph should be included.

Minor comments:

  • I suggest to the authors to include in the manuscript title the origin of EVs, i.e. human mesenchymal stromal cells
  • In the Materials and Methods section, the authors omitted the CD63 antibody and wrote about a densitometric analysis which was not shown; this information should be included in the manuscript. Also, they should include the primer sequences of Rplp0.
  • Line 341, EVs is repeated twice.

Round 2

Reviewer 2 Report

Thank you, the authors have answered my requests. I encourage the authors to a future in-depth study of the mechanism of action.